

# A 2D image 3D reconstruction function adaptive denoising algorithm

Feng Wang, Weichuan Ni, Shaojiang Liu, Zhiming Xu, Zemin Qiu and Zhiping Wan

Guangzhou Xinhua University, Dongguan, Guangdong, China

## ABSTRACT

To address the issue of image denoising algorithms blurring image details during the denoising process, we propose an adaptive denoising algorithm for the 3D reconstruction of 2D images. This algorithm takes into account the inherent visual characteristics of human eyes and divides the image into regions based on the entropy value of each region. The background region is subject to threshold denoising, while the target region undergoes processing using an adversarial generative network. This network effectively handles 2D target images with noise and generates a 3D model of the target. The proposed algorithm aims to enhance the noise immunity of 2D images during the 3D reconstruction process and ensure that the constructed 3D target model better preserves the original image's detailed information. Through experimental testing on 2D images and real pedestrian videos contaminated with noise, our algorithm demonstrates stable preservation of image details. The reconstruction effect is evaluated in terms of noise reduction and the fidelity of the 3D model to the original target. The results show an average noise reduction exceeding 95% while effectively retaining most of the target's feature information in the original image. In summary, our proposed adaptive denoising algorithm improves the 3D reconstruction process by preserving image details that are often compromised by conventional denoising techniques. This has significant implications for enhancing image quality and maintaining target information fidelity in 3D models, providing a promising approach for addressing the challenges associated with noise reduction in 2D images during 3D reconstruction.

# INTRODUCTION

Deep learning techniques have made significant advancements in various fields such as image processing, natural language processing, and network security detection. These techniques have shown promising results in experiments, exhibiting low error rates during training and strong generalization capabilities for test data. However, noisy images in image processing and noise in other types of data can negatively impact the accuracy of deep learning algorithms (*Tibi et al., 2021*; *Giannatou et al., 2019*; *Zhang et al., 2021*; *Ye, Li & Chen, 2021*; *Singh, Mittal & Aggarwal, 2020*; *Hales, Pfeuffer & Clark, 2020*; *Zhu et al., 2022*). For instance, noise in speech recognition can lead to reduced accuracy in semantic prediction. The ubiquity of noise presents challenges in training deep learning algorithms, as

Corresponding author
Feng Wang, iswf@xhsysu.edu.cn

it is difficult to collect pure data for training purposes. Even if the activity being studied is not affected by noise, real-world applications may introduce noisy data due to the environment, which can significantly impact accuracy in detection and processing tasks. There are three main categories of denoising algorithms: spatial domain-based, transform domain-based, and learning-based algorithms. Each category has its own advantages and disadvantages (*Li et al., 2022a*; *Pimpalkhute et al., 2021*; *Yan et al., 2021*; *Kazuaki et al., 2022*; *Zhang et al., 2021*). Spatial domain-based algorithms are easy to understand and implement, but they may not perform well in removing strong noise. Transform domain-based algorithms are more effective in handling various types of noise, but they require experience and professional knowledge during processing. Learning-based algorithms can learn data relationships better, but they typically require a large amount of labeled data for training and may suffer from overfitting issues. In addition, the 3D reconstruction of 2D images is a highly researched topic in computer vision and image processing. Modeling techniques can help explain changes in natural images, enabling neural networks to better understand image details. These techniques have practical applications in various computer vision-related fields. For instance, in autonomous driving systems, converting 2D camera images into 3D can help estimate scene depth. In the medical imaging field, it can assist with on-site diagnosis and simulation training (*Gao & Yuille, 2017*; *Sisniega et al., 2021*; *Li et al., 2022a*; *Zhang, Cui & Ding, 2021*; *Sun, 2021*; *Yu et al., 2021*; *Svahn et al., 2021*; *Wu et al., 2021*). However, noise in 2D images can pose challenges in the 3D reconstruction process. It may result in incomplete reconstruction of target details or even mistaken noise points for features, adversely affecting target recognition. This highlights the need for effective denoising algorithms to improve the accuracy and quality of 3D reconstructions.

In this study, an adaptive denoising algorithm for the 3D reconstruction function of 2D images is proposed, utilizing Generative Adversarial Networks (GANs) as a neural network model trained on adversarial learning data. The main objective is to generate noise-free images realistically, even in the presence of noise. The project involves preparing a noise-free image generator using GANs, which reproduces the image accurately despite the presence of noise. In this approach, noise generators are introduced and trained using the noise-free image generator as a reference. Distribution and transformation constraints are incorporated into the noise generator function, guiding it to capture specific noise components effectively. This ensures that the method can adaptively learn the noise-free image generator, even when training images contain significant amounts of noise. To preserve target information effectively, this study constructs a 3D model of the target region. A two-dimensional monocular image is utilized as input, combined with target information, and a confidence factor is introduced to improve the preservation of target details from the original image. Through experimental evaluations, the algorithm demonstrates stable preservation of image details. The reconstruction effect is tested on noisy 2D images, and the reconstruction of the 3D model is tested on noisy pedestrian datasets. The results show an average noise reduction rate exceeding 95%, with the 3D model effectively retaining most of the feature information from the original image. In summary, this study proposes an adaptive denoising algorithm for 3D reconstruction using GANs. By effectively reducing noise and preserving target details, the algorithm

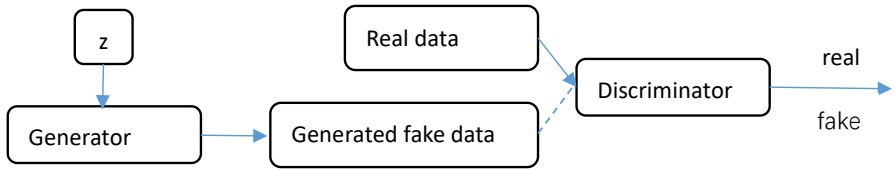

**Figure 1** **GAN structure.** GAN is a deep learning model consisting of two parts: the generator, which is used to generate new data similar to the given data, and the discriminator, which is used to judge the authenticity of the data generated by the generator. The two parts interact with each other through training, allowing the generator to continuously learn the skills of generating real data, and improving the generator's generation ability.

offers promising results in improving the quality and accuracy of image reconstruction. It introduces innovative techniques to handle noise and enhance the fidelity of reconstructed 3D models from 2D images.

## GENERATING ADVERSARIAL NETWORKS

Goodfellow proposed the Generative Adversarial Network (GAN), which employs two convolutional neural networks in a game-based training approach to generate images resembling the original picture (_Ozkanoglu & Ozer, 2022_; _He, Wandt & Rhodin, 2022_; _Iqbal & Ali, 2018_; _Kumar et al., 2022_; _Zhao, Wei & Wong, 2022_). The GAN model consists of two essential components: the generator and the discriminator.

In Fig. 1, the structure of the generative adversarial network is depicted. The generator is responsible for generating new data that is similar to the given data. Its objective is to create samples with high resemblance to real data. On the other hand, the discriminator's role is to determine the authenticity of the data generated by the generator. It aims to differentiate between real and generated samples accurately. During the training process, the generator and discriminator interact with each other, creating a competitive feedback loop. This allows the generator to progressively improve its ability to generate data that appears genuine, while the discriminator becomes more skilled at distinguishing between real and generated data. The GAN model facilitates the generator in learning the skills necessary to produce high-quality and realistic data, while also enhancing its generation capability. This interplay between the generator and discriminator enables the GAN to generate novel, authentic-like data that closely resembles the original picture.

GAN offers several key advantages, including:

High-quality generation: GAN's generator produces samples of superior quality that closely resemble actual data. This enables GAN to generate highly realistic and even imaginative data, making it applicable in various practical scenarios.

Scalability: GAN can be trained using different types of data, such as images, text, audio, video, and 3D models. This makes it versatile and applicable across multiple fields, providing flexibility for various applications.

Unsupervised learning: GAN operates through unsupervised learning, meaning it doesn't require labeled data for training. This makes it more universal since it can work with unlabeled datasets without the need for extensive data labeling.

Ability to learn data distribution: GAN can simulate and learn the underlying distribution of the data it processes. This capability is valuable for data reconstruction and image generation tasks. The generator in GAN not only generates new samples but can also predict new data distributions, which is crucial in the field of data science.

High effectiveness and training efficiency: GAN training is highly efficient as the generator and discriminator are trained simultaneously in a competitive manner. This allows GAN to converge rapidly, leading to efficient and effective training processes.

## ARTICLE ALGORITHM

### Region noise reduction

In this study, we incorporate the intrinsic visual characteristics of the human eye, taking into account the entropy value of the image, which accurately reflects the image signal (*Pulgar et al., 2021*; *Zheng et al., 2021*). However, in practical computer processing, calculating the entropy value can be computationally intensive. Therefore, we propose a simplified calculation process as follows.

Assume that the information level of the original image is $L$. The number of targets with the information $i$ is $ni$. The total number of pixels of the image is $N$. The probability of occurrence of each target can be obtained as $Pi$. then we have $Pi = ni/N$. In the image segmentation algorithm, a threshold $\lambda$ is used to classify the image information level into two classes. The target class $Co$ and the background class $Cb$. The target part $f(x, y) \geq \lambda$ and the background part $f(x, y) < \lambda$. Thus, the image is effectively segmented into subsets that do not overlap. Thus, the ratio of its target to background occurrences is:

$$P_{\mathrm{b}} = \sum_{i=0}^{\lambda} P_i \text{ and } P_{\mathrm{f}} = \sum_{i=\lambda+1}^{L-1} P_i \qquad (1)$$

Target mean: $\quad \mu_{\mathrm{b}}(t) = \dfrac{\sum_{i=0}^{\lambda} i P_i}{P_{\mathrm{b}}(t)} \qquad (2)$

Background mean : $\quad \mu_{\mathrm{f}}(t) = \dfrac{\sum_{i=\lambda+1}^{L-1} i P_i}{P_{\mathrm{f}}(t)} \qquad (3)$

The average value of the information in the whole image is: $\mu = \sum_{i=0}^{L-1} i P_i$

Therefore, the inter-class variance of the image is obtained according to the inter-class variance formula:

$$\sigma_{\mathrm{B}}^2(t) = P_{\mathrm{b}}(t)[\mu_{\mathrm{b}}(t) - \mu]^2 + P_{\mathrm{f}}(t)[\mu_{\mathrm{f}}(t) - \mu]^2 \qquad (4)$$

Simplifying, we get: $\sigma_{\mathrm{B}}^2(t) = P_{\mathrm{b}}(t)[1 - P_{\mathrm{b}}(t)][\mu_{\mathrm{b}}(t) - \mu_{\mathrm{f}}(t)]^2 \qquad (5)$

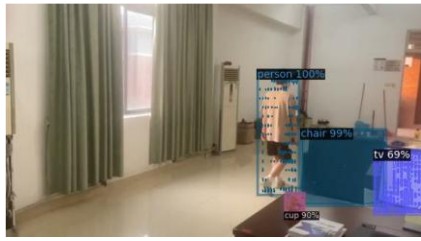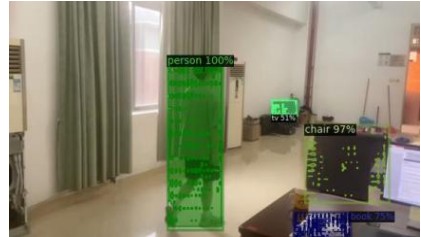

**Figure 2** Segmentation area map.

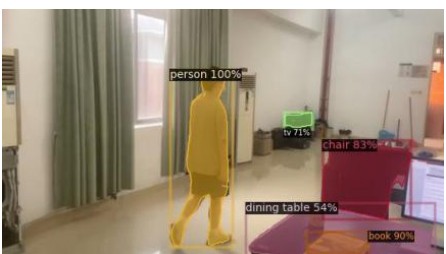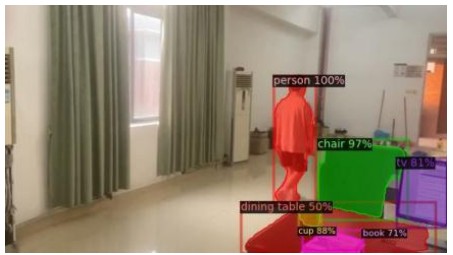

**Figure 3** Segmentation target map.

To make the article algorithm can better cope with different background images. In this article, the background of its image is analyzed for complexity. Suppose $H_{ij}$ is the local neighbourhood entropy centred on $(i, j)$. The expression of its function is shown as follows:

Local neighbourhood entropy: $H_{ij} = -\sum_{i=1}^{m}\sum_{j=1}^{n} P_{ij} \lg P_{ij}$ (6)

Where: $m \times n$ is the size of a local neighbourhood. $P_{ij}$ is the probability of the target distribution at point $(i, j)$.

Figures 2 and 3 are taken as samples. In order to better reflect the complexity of the image background. In this article, the neighbourhood entropy is converted into a background factor between $(0, 1)$ by establishing an affiliation function, which is shown as follows:

The affiliation function of the background factor: $K_{ij} = \dfrac{H_{ij} - H_{min}}{Hmin_{max}}$ (7)

The background factor of the image background is finally obtained, and the target area is finally targeted.

## Noise reduction processing
### Background area processing
Let the image signal processed for noise reduction be $I = f + n$, where f is the image signal. $n$ is the noise signal. In this article, the noise reduction function of the image is obtained

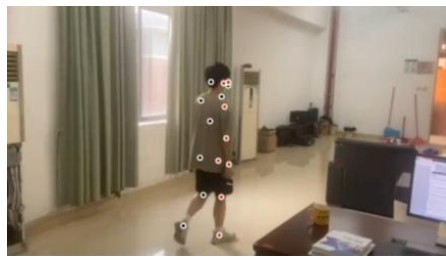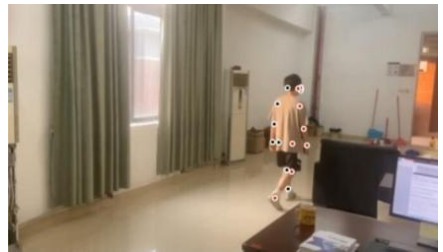

**Figure 4**  **Target key point extraction.**

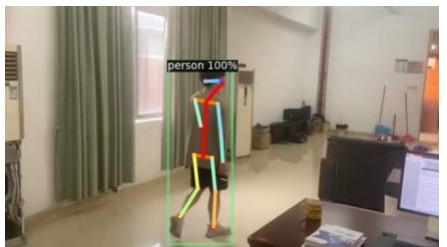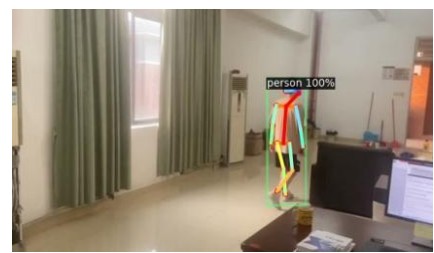

**Figure 5**  **2D target pose extraction.**

by improving the soft threshold denoising algorithm. The expression of the function is defined as follows:

$$\hat{C}_{ij} = \begin{cases} \mu \cdot \text{sgn}(C_{i,j})(|C_{i,j}| - \lambda) & , |C_{ij}| \geq \lambda \\ 0 & |C_{ij}| < \lambda \end{cases} \qquad (8)$$

Where $\text{sgn}(n) = \begin{cases} 1 & ,n > 0 \\ -1 & ,n \leq 0 \end{cases}$. $\mu$ is the weight constant. $\lambda$ Is the threshold value. Although the selection of the threshold value directly affects the noise reduction process of the image. Considering the region's more practical information, it is easy to cause the "overkill" phenomenon. In this article, the weight $\mu$ is defined. The value of $\mu$ is 0.6 and is substituted into the image noise reduction formula. Then the formula is as follows:

$$\hat{C}_{ij} = \begin{cases} 0.6\text{sgn}(C_{i,j})(|C_{i,j}| - \lambda) & , |C_{ij}| \geq \lambda \\ 0 & |C_{ij}| < \lambda \end{cases} \qquad (9)$$

### Target area processing

*Confidence processing.*  This study proposes a method to introduce a confidence factor in GAN. The confidence data is obtained by combining the data generated by the generator with the data in the discriminator.

As shown in Figs. 4 and 5, the ground truth of $S$ is calculated from the 2D points $x_{j,k}$ and $x_{i,k}$ annotated in the image. where $x_{j_1,k}$ and $x_{j_2,k}$ denote the two key points $j_1$ and $j_2$ corresponding to the real pixel points of a person $k$ in the figure.

If a pixel point $C_{real}$ is located on this target node. $L^*_{c,k}(C_{real})$ the valued table is a unit vector from a key point $j_1$ to key point $j_2$. The corresponding vector is a zero vector for pixel points not on the torso. Then the values of $L^*_{c,k}(C_{real})$ Are as follows:

$$L^*_{c,k}(C_{real}) = \begin{cases} v & if\ C_{real} \quad on\ \ c,k \\ 0 & otherwise \end{cases} \tag{10}$$

where $v = \frac{(x_{j_2,k} - x_{j_1,k})}{|x_{j_2,k} - x_{j_1,k}|_2}$ Denotes the unit direction vector corresponding to this torso.

The target pixel point satisfies the following function:

$$0 \leq v\left(C_{real} - x_{j_1,k}\right) \leq l_{c,k} and |v_\perp(C_{real} - x_{j_1,k})| \leq \sigma_l \tag{11}$$

where the inner table $\sigma_l$ shows the distance between pixel points. The torso length is $l_{c,k} = |x_{j_2,k} - x_{j_1,k}|_2$ and $v_\perp$ Denotes the vector perpendicular to $v$.

### Generator and discriminator

We optimize the loss $L_{MSE}$ of the generator and the adversarial loss $L_{adv}$ of the discriminator.

$$L_{MSE} = \sum_{i=1}^{N} \sum_{j=1}^{M} (C_{ij} - \hat{C}_{ij})^2 \tag{12}$$

$$L_{adv} = \sum_{i=1}^{N} (\hat{C}_j - D(\hat{C}_j, X))^2 \tag{13}$$

$$L_G = L_{MSE} + \lambda L_{adv} \tag{14}$$

The primary objective of the discriminator is to discern whether a given heat map is real or fake, generated by the generator (*Vo et al., 2021*; *Dong et al., 2021*; *Li et al., 2022b*; *Lu & Su, 2021*; *Luo et al., 2021*). To accomplish this, we optimize the loss function of the discriminator to improve its ability to differentiate between real and fake heat maps. The optimization process aims to enhance the discriminator's discriminative capabilities and make it more effective in accurately identifying the authenticity of the input heat maps. By fine-tuning the loss function, we aim to facilitate the discriminator in becoming increasingly proficient at recognizing real and generated heat maps.

$$L_{real} = \sum_{j=1}^{N} (C_j - D(C_j, X))^2 \tag{15}$$

$$L_{noiseless} = \sum_{j=1}^{N} (\hat{C}_j - D(\hat{C}_j, X))^2 \tag{16}$$

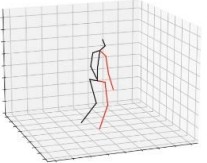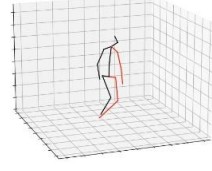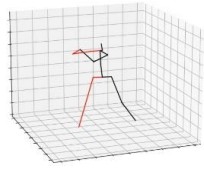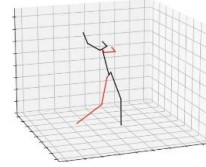

**Figure 6  3D target pose extraction.**

$$L_G = L_{real} + k_t L_{\text{noiseless}} \tag{17}$$

$$k_{t+1} = k_t + \lambda_k (L_{real} - L_{\text{noiseless}}). \tag{18}$$

The $kt$ in the above equation is used to constrain the capability of the resolver.

As shown in Fig. 6, it can be seen that all components in this network, including the confidence map, are learned from the image only. In processing this deep neural network, the feature factors are derived from the original image. These feature factors are mapped to the image with depth information in a recombination manner to construct a new image.

If the expected target is irrelevant for noisy samples, the pose in this image is not credible for the body construction. Therefore, if the information is closer to the target information, the corresponding vector is 1, and vice versa, the corresponding vector is 0. The formula is as follows:

$$C_{real} = \begin{cases} 1 & if \, ||K_{ij}|| < \tau \\ 0 & if \, ||K_{ij}|| \geq \tau \end{cases} \tag{19}$$

$K_{ij}$ represents the value of the affiliation function for the background factor.

The system framework is shown in Fig. 7.

## SIMULATION

In this study, the experimental platform utilized Ubuntu 18.04 as the operating system. Anaconda was utilized as the software platform, while PyTorch served as the deep learning framework. In order to assess the practicality of the algorithm, real pedestrian videos captured by the researchers were used, with noise intentionally added to the images. The model was trained and tested on a high-performance server equipped with a 2080 × 4 GPU. A GAN-based image generator, capable of producing noise-free images, was trained and evaluated using the noisy dataset. The noise removal rate was measured, and comparable experiments were conducted with other algorithms under the same experimental conditions. Data comparison was performed to evaluate the performance of the proposed approach against the comparison algorithms.

In order to validate the denoising effectiveness of the algorithms proposed in this article, several traditional and literature algorithms are selected for comparison (*Chen & Han,*

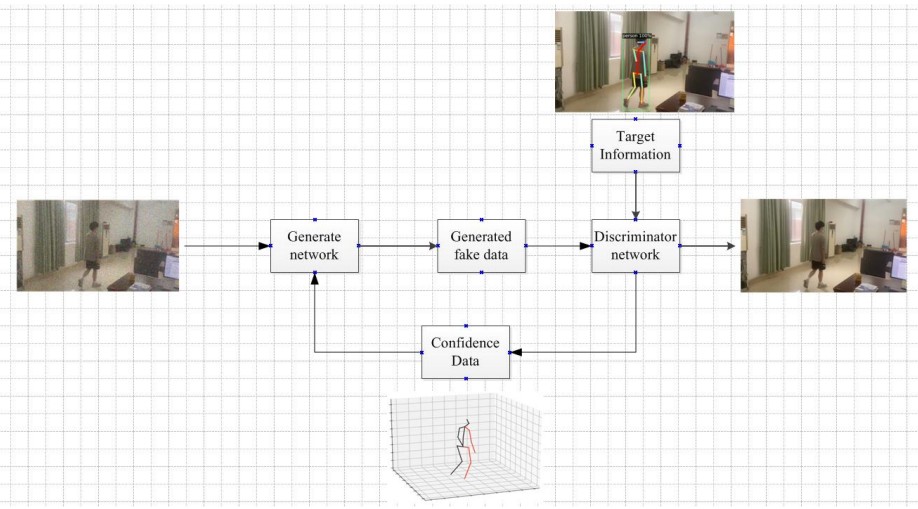

Figure 7    Framework diagram of the algorithm.

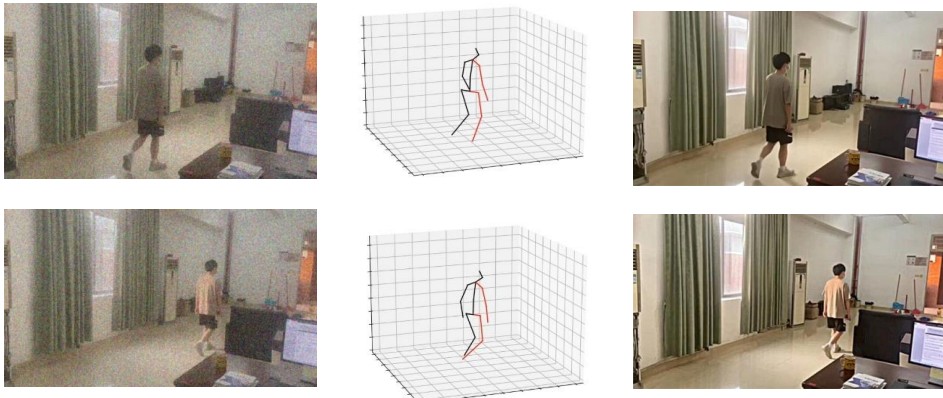

Figure 8    3D model and denoising effect of the algorithm.

*2005*; *Hu et al., 2021*; *Deng et al., 2020*; *Frazier-Logue & José Hanson, 2020*). The simulation process involves collecting data for each algorithm and comparing them. The equations used for comparison are based on observing the signal-to-noise ratio (PSNR) values of each algorithm.

$$\text{Signal-to-noise ratio: } PSNR = 10\log\left(\frac{\sum_{i=1}^{N} x^2 i}{\sum_{i=1}^{N}(x[i]-\hat{x}[i])^2}\right). \tag{20}$$

As depicted in Fig. 8, it is noticeable that the denoising algorithm proposed in this study effectively eliminates noise while preserving the edge and texture detail features

**Table 1  Data table of PSNR values for each denoising algorithm.** Table 1 shows this algorithm outperforms other denoising algorithms regarding PSNR values under different noise factors. It offers better performance than other denoising techniques

| $\sigma$ | PSNR dB | | | | | |
|---|---|---|---|---|---|---|
| | Noisy images | Traditional algorithms | Literature algorithm 1 | Literature algorithm 2 | Literature algorithm 3 | Article algorithm |
| 10 | 26.0 | 29.6 | 31.8 | 35.8 | 38.2 | 40.1 |
| 20 | 24.1 | 27.1 | 27.9 | 34.8 | 38.2 | 38.6 |
| 30 | 22.1 | 25.9 | 27.5 | 34.1 | 34.7 | 38.1 |
| 40 | 18.3 | 25.8 | 26.5 | 32.1 | 33.3 | 34.0 |
| 50 | 15.8 | 25.2 | 23.8 | 31.9 | 30.8 | 32.8 |

present in the original image. The resulting image exhibits high visual quality, indicating the algorithm's ability to retain crucial visual information despite noise removal.

The objective of this article is to validate the effectiveness of the proposed method, particularly in terms of successfully denoising images while preserving target information. The superiority of the algorithms is assessed by comparing their PSNR values and time consumption. Through the experiments, the table shows the comparative PSNR values of the results obtained by different denoising algorithms.

Table 1 demonstrates that the proposed algorithm outperforms other denoising algorithms in terms of PSNR values across various noise factors. It consistently exhibits better performance compared to alternative denoising techniques. The algorithm showcases a significant difference of up to 3.6 dB compared to the literature algorithm 3, certifying its reliability. The evaluation index performance is particularly commendable at a noise variance of $\sigma = 50$. Furthermore, assessing the image metrics at different variances reveals a decrease in PSNR values as the noise variance increases. Despite this decrease in performance, the proposed algorithm still maintains a superior denoising effect compared to other algorithms. This comprehensive investigation effectively substantiates the image fidelity achieved by the proposed algorithm.

In the process of denoising, it is inevitable that some image information may be lost and residual noise information may still remain. Consequently, the grayscale values of certain pixels in the image may change accordingly. This presents an opportunity to evaluate and compare the denoising effects of various methods based on the grayscale histogram of the image.

Figure 9A displays the original image along with its grayscale histogram, while Fig. 9B depicts the grayscale histogram of the image after adding noise. By comparing the histograms of different denoising algorithms, it can be observed that the histograms of the literature algorithm 3 and the proposed algorithm in this article closely resemble the histogram of the original image. Furthermore, among these algorithms, the proposed algorithm in this article preserves the image details to the highest extent. This indicates that the proposed algorithm achieves the most effective denoising results, as it successfully retains the original image's characteristics and minimizes the impact of noise on the image.

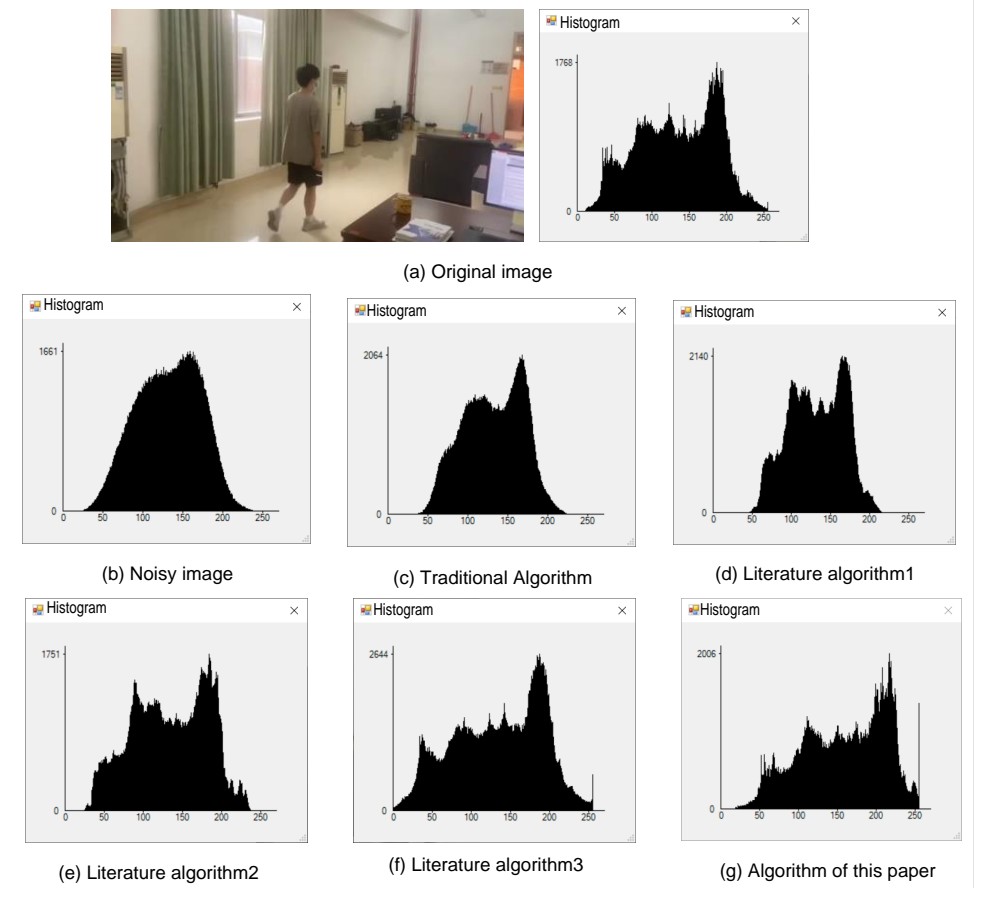

**Figure 9** **Histogram of each algorithm.** (A) Original image; (B) noisy image; (C) traditional algorithm; (D) literature algorithm 1; (E) literature algorithm 2; (F) literature algorithm 3; (G) algorithm of this article.

Thus, based on the comparison of histograms, it can be concluded that the algorithm proposed in this article offers the most favorable denoising effect.

In order to verify the effectiveness of the proposed algorithm in denoising and preserving target information in images, a comparative experiment is conducted. The experiment focuses on advanced denoising of targeted image information. The results of this comparison experiment are presented in Fig. 10, showcasing the denoising effects achieved by the different algorithms.

Through a comparison of the denoising effects achieved by different algorithms, it is evident that the algorithm proposed in this article is capable of preserving the target signal of the image while effectively denoising the noisy image. This results in a visually superior outcome with a more precise overall impact compared to the original image. Upon closer inspection of the key details in the denoising effect of each algorithm, it is apparent that the algorithm proposed in this article holds a significant advantage in preserving target details. The denoising results align with the intended experimental objectives, validating the effectiveness of the proposed algorithm. While the literature algorithm 3 can achieve

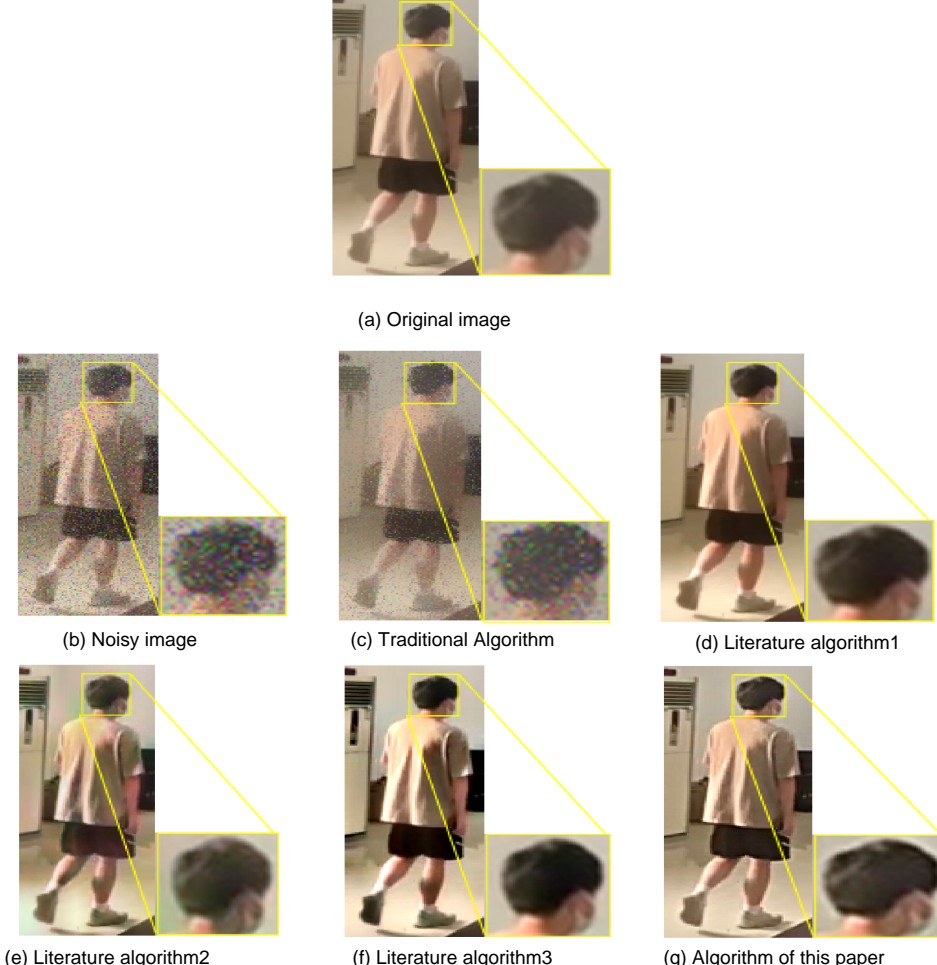

(a) Original image

(b) Noisy image   (c) Traditional Algorithm   (d) Literature algorithm1

(e) Literature algorithm2   (f) Literature algorithm3   (g) Algorithm of this paper

**Figure 10   Comparison of denoising details of each algorithm.** (A) Original image; (B) noisy image; (C) traditional algorithm; (D) literature algorithm 1; (E) literature algorithm 2; (F) literature algorithm 3; (G) algorithm of this article.

a similar effect to the original image, it often leads to an excessively subdued background. In contrast, the algorithm proposed in this article presents better denoising results for both the target and the background, with the clearest details. This effectively showcases the strengths of the algorithm proposed in this article in the field of denoising.

This article also highlights the efficiency of the proposed method in eliminating image noise. The favorable denoising performance can be primarily attributed to the utilization of GAN within the optimization algorithm. The time efficiency of the proposed method is demonstrated in Fig. 11. It is evident that the algorithm proposed in this article maintains a good balance between denoising performance and time efficiency. Although there is a slight increase in denoising efficiency compared to the literature algorithm, the algorithm in this article operates with a time efficiency difference of 0.153s. This signifies that the

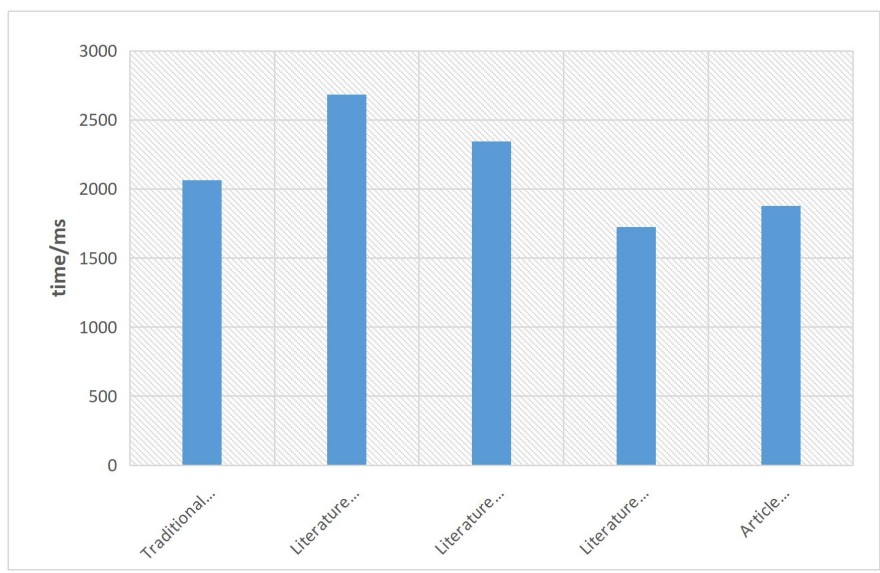

**Figure 11**   **Denoising efficiency graph of each algorithm.**

proposed algorithm effectively achieves better denoising results while still maintaining a reasonable level of computational efficiency.

## CONCLUSION

We proposed the design of a noise-free image generator based on GAN, which is a neural network trained using adversarial learning to model data distribution. The primary objective of this project was to develop a noise-free image generator that can produce high-fidelity images despite the presence of noise. To achieve this, we combined the inherent visual properties of the human eye with the entropy value of the input image to divide it into different regions. The background region was subjected to threshold denoising, while the target region undergoes specific processing. Additionally, we constructed a 3D model for the target using an adversarial generation network, starting from the 2D target image with noise. By combining the target information with a confidence factor, we aimed to better preserve the target's detailed information from the original image. The experimental results demonstrate the algorithm's ability to efficiently denoise noisy images while retaining the detail signal of the target image, which aligns with the expected outcomes. However, it is important to note that the adversarial generative network often involves complex calculations and requires tedious parameter adjustments, potentially impacting the training timeliness. In future works, we aim to address these limitations by exploring high-speed and effective algorithms that can offer improved efficiency without compromising on the denoising accuracy achieved by the proposed method.

### Funding

This research is supported by the Guangzhou Science and Technology Project, No 202201011731, the Guangdong Key Discipline Scientific Research Capability Improvement Project with No 2021ZDJS144, the Research Fund for Faculty Members of Guangzhou Xinhua University Project, No 2020KYYB03, and the Guangdong Key Discipline Scientific Research Capability Improvement Project, No 2022ZDJS151. The funders had no role in study design, data collection and analysis, decision to publish, or preparation of the manuscript.

### Grant Disclosures

The following grant information was disclosed by the authors:
The Guangzhou Science and Technology Project: 202201011731.
Guangdong Key Discipline Scientific Research Capability Improvement Project: 2021ZDJS144.
Research Fund for Faculty Members of Guangzhou Xinhua University Project: 2020KYYB03.
Guangdong Key Discipline Scientific Research Capability Improvement Project: 2022ZDJS151.

### Competing Interests

The authors declare there are no competing interests.

### Author Contributions

- Feng Wang conceived and designed the experiments, performed the experiments, analyzed the data, prepared figures and/or tables, authored or reviewed drafts of the article, and approved the final draft.
- Weichuan Ni conceived and designed the experiments, performed the experiments, performed the computation work, prepared figures and/or tables, and approved the final draft.
- Shaojiang Liu conceived and designed the experiments, performed the computation work, prepared figures and/or tables, and approved the final draft.
- Zhiming Xu analyzed the data, performed the computation work, prepared figures and/or tables, and approved the final draft.
- Zemin Qiu performed the experiments, analyzed the data, performed the computation work, authored or reviewed drafts of the article, and approved the final draft.
- Zhiping Wan performed the experiments, analyzed the data, authored or reviewed drafts of the article, and approved the final draft.

### Data Availability

The relevant data is available in the Supplementary Files.

## Supplemental Information

Supplemental information for this article can be found online at http://dx.doi.org/10.7717/peerj-cs.1604#supplemental-information.

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
