# Peer review of "A 2D image 3D reconstruction function adaptive denoising algorithm"

_PeerJ Computer Science, doi:10.7717/peerj-cs.1604_

## Round 0.1 · original submission · Major Revisions

Dear Authors,
Experts in the field have reviewed your paper, and you will see they have a couple of revisions to address. Please take into account those specific improvement areas along with my comments below.

AE Comments: Please improve the language of the manuscript.
2. Improve the quality of the figures eg figure 7
3. make the abstract more concise

Reviewer 1 ·

Basic reporting

In this study, a noise-free image generator is used to duplicate pictures using GANs even in the presence of noise. While the target region is treated, the background region is denoised using thresholding. An adversarial generative network is used to create a three-dimensional model of a two-dimensional target picture with noise. The input is a two-dimensional monocular picture, and the target information is coupled with a confidence coefficient to maintain the target features from the original image. This technique effectively denoises noisy photos while preserving the target image's fine features.

The article's organisation and structure must be revised. Some of the current titles, particularly in Section 3, may be inappropriate or irrelevant to the content. Consider altering the names to better fit the substance and flow of the information in the sections.

It is suggested that the model and results sections be supplemented with relevant subheadings to give a clear classification and improve the readability and organisation of the paper.

For certain details, use tables sparingly and make sure the data supplied in them does not duplicate results described elsewhere in the text.

Vertical rules and shading in table cells should be avoided. It is also proposed that in the Conclusion, the important results be summarised one by one with the marks of 1. 2. 3. or i. ii. iii. etc.
Most sentences are incomplete or include grammatical and/or spelling errors.

Experimental design

The current flaws in the abstract are not highlighted sufficiently, and the presentation of the optimised approach is insufficiently thorough. A brief explanation of the methods to be used by the author should also be included.

The suggested aims, uniqueness, and holes to be addressed should be more fully justified in the final section of the introduction.

For example, in this sentence "in image processing, noisy images can affect the object classification accuracy of deep learning algorithms". The author should elaborate on how image noise affects object classification accuracy.

Validity of the findings

Although GANs are widely used models for image data training and generation, it is unclear how this research improves on previous methodologies. The text should explicitly explain the unique improvements or advances made in this study compared to earlier publications.

Cite this review as

Reviewer 2 ·

Basic reporting

In this paper, a 3D model of the target region is constructed. The 2D monocular image is taken as input, combined with the object information, and a confidence coefficient is introduced. In turn, the target details of the original image are better preserved. Through experiments, the algorithm can ensure that the details of the image have good stability. However, it is not acceptable for publication in its present form. Please carefully address the issues raised in the comments.

(1) The article as concise as possible compression, can briefly introduce the background principle. In addition, the algorithm proposed in this article should be emphasized, and other algorithms can be briefly introduced and compared to highlight the key points;
(2) The purpose of literature citations is to summarize existing research findings and support one's own arguments. The authors should compare the research content of different references and avoid using overly general expressions;
(3) One prominent issue is the presence of a significant amount of repetitive statements throughout the manuscript (in the introduction and abstract);
(4) The structure and content of this paper should be readjusted at the end of the introduction. This place is best expressed in another way;
(5) Section 2 should be merged with the Introduction, establishing a strong connection between GANs and the target problem under study;
(6) Regarding the claim that this research can enhance the noise resistance of two-dimensional images in the process of three-dimensional reconstruction and better preserve the target details from the original image, I could not find a corresponding section in the manuscript that explains the process of three-dimensional reconstruction;
(7) Discussion section needs to be a coherent and cohesive set of arguments that take us beyond this study in particular, and help us see the relevance of what the authors have proposed;
(8) The conclusion provides a good summary of the paper's contributions and future research directions. However, the authors could benefit from more explicitly highlighting the novelty and potential impact of their proposed method;
(9) "The sample size rationale and its associated implications should be expounded upon with greater clarity, emphasizing potential generalizability limitations."

Experimental design

See above

Validity of the findings

See above

Additional comments

See above

Cite this review as

---

## Round 0.2 · accepted · Accept

Based on the input from the experts in the field. I am pleased to inform you that they are satisfied with the current version of the paper and therefore, it is being recommended for publication.

Reviewer 2 ·

Basic reporting

The paper has been well revised according to my comments and now it looks good to accept, therefore I recommend it for acceptance.

Experimental design

The paper has been well revised according to my comments and now it looks good to accept, therefore I recommend it for acceptance.

Validity of the findings

The paper has been well revised according to my comments and now it looks good to accept, therefore I recommend it for acceptance.

Additional comments

The paper has been well revised according to my comments and now it looks good to accept, therefore I recommend it for acceptance.

Cite this review as